# Fusogenic Liposomes for the Intracellular Delivery of Phosphocreatine

**DOI:** 10.3390/ph17101351

**Published:** 2024-10-10

**Authors:** Okhil K. Nag, Eunkeu Oh, James B. Delehanty

**Affiliations:** 1Center for Bio/Molecular Science and Engineering Code 6900, US Naval Research Laboratory, Washington, DC 20375, USA; 2Optical Sciences Division, Code 5600, US Naval Research Laboratory, Washington, DC 20375, USA; eunkeu.oh2.civ@us.navy.mil

**Keywords:** fusogenic liposomes, phosphocreatine, adenosine triphosphate, plasma membrane

## Abstract

**Background/Objective:** Maintaining intracellular adenosine triphosphate (ATP) levels is essential for numerous cellular functions, including energy metabolism, muscle contraction, and nerve impulse transmission. ATP is primarily synthesized in mitochondria through oxidative phosphorylation. It is also generated in the cytosol under anaerobic conditions using phosphocreatine (PCr) as a phosphate donor to adenosine diphosphate. However, the intracellular delivery of exogenous PCr is challenging as it does not readily cross the plasma membrane. This complicates the use of PCr as a therapeutic agent to maintain energy homeostasis or to treat conditions like cerebral creatine deficiency syndrome (CDS), which results from defective creatine transporters. **Methods:** This study employs the use of fusogenic liposomes to deliver PCr directly into the cytosol, bypassing membrane impermeability issues. We engineered various 1,2-dioleoyl-3-trimethylammonium-propane (DOTAP)-based fusogenic liposomes, incorporating phospholipids such as 1,2-dioleoyl-sn-glycero-3-phosphoethanolamine (DOPE) in combination with phospholipid-aromatic dye components to facilitate membrane fusion and to enhance the delivery of the PCr cargo. Liposomal formulations were co-loaded with membrane-impermeable chromophores and PCr and studied on live cells using confocal microscopy. **Conclusions:** We demonstrated the successful intracellular delivery of these agents and observed a 23% increase in intracellular ATP levels in cells treated with PCr-loaded liposomes. This increase was not observed with free PCr, confirming the effectiveness of the liposome-based delivery system. Additionally, cell viability assays showed minimal toxicity from the liposomes. Our results indicate that fusogenic liposomes are a promising method for the delivery of PCr (and potentially other cell-impermeable therapeutic agents) to the cellular cytosol. The approach demonstrated here could be advantageous for treating energy-related disorders and improving cellular energy homeostasis.

## 1. Introduction

Adenosine triphosphate (ATP) is the energy source for use and storage at the cellular level. It plays critical roles in various cellular processes, including biochemical transformations and syntheses, muscle contraction, and nerve impulse propagation. ATP is mainly generated in cells via the oxidative phosphorylation of adenosine diphosphate (ADP) in the mitochondria during aerobic glucose catabolism. ATP is also generated from ADP in the cytosol under anaerobic conditions using phosphocreatine (PCr) as an intermediate phosphate group source [1]. PCr is generated in the mitochondria via the phosphorylation of creatine (Cr) using ATP as a phosphate donor in a reaction catalyzed by mitochondrial creatine kinase (CK). Mitochondrial PCr then diffuses into the cytosol, where it is converted into Cr through the transfer of its high-energy phosphate group to ADP with the help of cytosolic isoforms of CK to generate ATP. The cytosolic Cr returns to the mitochondria to recycle the process. This ATP regeneration process via PCr typically occurs within seconds of intense muscular or neuronal effort, acting as a quick and accessible reserve of high-energy phosphates for the recycling of ATP in muscle tissues and the brain.

Cr plays critical roles in the mitochondrial and cytoplasmic ATP recycling process, and its intracellular transport is mediated by a specific plasma membrane-resident transporter (CRT1/SLC6A8) that belongs to a family of solute carrier 6 (SLC6) proteins [2]. CRT1 translocates Cr across tissue barriers, including the plasma membrane, into targeted cells such as neurons and muscle cells. Individuals lacking CRT1 or harboring mutations in the CRT1 gene develop creatine transporter deficiency (CTD), a primary cause of cerebral creatine deficiency syndrome (CDS). CTD disrupts cellular ATP generation, leading to an array of clinical manifestations, including severe intellectual disability, epilepsy, autism, development delay, and motor dysfunction. In such situations, the intracellular delivery of exogenous PCr could be a therapeutic option for treating CDS and maintaining energy homeostasis. However, PCr does not readily cross the plasma membrane into cells. Additionally, unlike Cr, there is no protein-mediated transport of PCr. The generation and consumption of PCr are part of an intracellular process. Due to its charged nature, the PCr delivered from the extracellular bulk solution is not readily internalized into cells, and it primarily binds to the plasma membrane due to its amphipathic (or cationic) nature [3]. The ionic characteristics of PCr enable it to bind to the polar phospholipid heads of membranes, where it stabilizes the phospholipid bilayer and decreases both membrane fluidity and permeability. Thus, PCr is used for myocardial protection against reperfusion during cardiac surgery [3,4]. Furthermore, PCr administered intravenously undergoes hydrolysis, forming inorganic phosphate and Cr, resulting in a lack of ATP production when functional CRT1 is absent.

Currently, there are no reports on the direct intracellular delivery of PCr and its implications for ATP generation. An ideal approach to the delivery of PCr is one that would circumvent its interaction with the plasma membrane and deliver it directly into the cytosolic compartment of cells. Thus, our main goal in the work reported here was to develop fusogenic liposomes to deliver PCr to the cytosol of cells. Fusogenic liposomes comprise a predefined mix of phospholipids to enable the fusion of the liposomal membrane to the plasma membrane, resulting in the delivery of the aqueous liposomal contents into the cytosol. Fusogenic liposomes have been used to deliver various cargos, including proteins, nucleic acids, antimicrobial agents, and nanoparticles [5,6,7,8,9,10]. One of the key advantages of fusogenic liposome technology is the avoidance of endocytosis and the endosomal entrapment of the cargo. Kube et al. showed the intracellular delivery of different proteins, such as eGFP, and R-phycoerythrin, using a fusogenic liposomal formulation composed of 1,2-dioleoyl-sn-glycero-3-phosphoethanolamine (DOPE) and 1,2-dioleoyl-3-trimethylammonium-propane (DOTAP), where the latter lipid plays a significant role in the membrane fusion mechanism [7]. In an effort to decipher the interaction mechanism of DOTAP-based fusogenic liposome with the plasma membrane, Kolašinac et al. studied formulations of various combinations of cationic, nonionic, and aromatic (chromophore) lipids for their optimal fusogenic properties [6]. This work showed that mixing phospholipids bearing an aromatic head and a cationic DOTAP in the lipid composition significantly enhances the induction of liposome–plasma membrane fusion. Recently, DOTAP-based fusogenic stealth liposomes were formulated with the addition of zwitterionic 1,2-dioleoyl-snglycero-3-phosphocholine (DOPC) and antifouling 1,2-distearoyl-sn-glycero-3-phosphoethanolamine-*N*-[methoxy(polyethylene glycol)-2000] (DSPE-PEG) in the lipids mixture [11]. This stealth fusogenic liposome showed efficient luciferase gene transfection in vivo and improved the gene-editing efficiency of CRISPR/Cas9-based therapeutics in an in vivo hepatitis B virus (HBV) model. In addition to DOTAP, fusogenic liposomes are also formulated with other lipid compositions such as *L*-α-phosphatidylcholine, hydrogenated (Soy) (HSPC) and 2-diphytanoyl-sn-glycero-3-phosphoethanolamine (DPhPE) [12]. This formulation was reported for its targeted tumor cell-specific efficient delivery of a lipophilic pro-apoptotic drug, thapsigargin (Tg), to tumor cells (CT26) and colorectal tumors in the mouse model.

Here, we formulated a series of DOTAP-based fusogenic liposomes composed of DOTAP, DOPE, and aromatic lipids such as 1,2-dioleoyl-sn-glycero-3-phosphoethanolamine-N-(lissamine rhodamine B sulfonyl) (Rhod-PE) or 1,2-dioleoyl-sn-glycero-3-phosphoethanolamine-N-(7-nitro-2-1,3-benzoxadiazol-4-yl) (NBD-PE). These liposomal formulations were loaded with different combinations of the cell-impermeable chromophores calcein, 4′,6-diamidino-2-phenylindolenucleic acid (DAPI), and PCr into the aqueous core to characterize intracellular delivery via membrane fusion. Using these custom fusogenic liposomes, we demonstrated the intracellular delivery of the impermeable chromophores and PCr into the cells using live cell confocal microscopy. The successful delivery of the chromophores was directly captured by confocal imaging and the efficient delivery of PCr into the cells was indirectly confirmed through the increase in ATP level in the cells using an ATP-sensing fluorescent probe.

## 2. Results and Discussion

The delivery of exogenous PCr to the cellular cytosol poses a considerable challenge, given the propensity of PCr to interact with and stabilize plasma membrane phospholipids. The intracellular delivery of PCr with minimal direct interactions with the lipids in the plasma membrane could offer various advantages in maintaining cellular energy homeostasis. The use of fusogenic liposomes as a vehicle for such applications can (1) protect PCr from extracellular hydrolysis, (2) minimize PCr interaction with cellular lipids, and (3) directly deliver PCr into the cytosol while avoiding other internalization pathways, such as endocytosis, when using other types of delivery vehicles (e.g., non-fusogenic liposomes). Thus, the goal of this study was to assess the delivery of PCr to the cytosol while maintaining its activity to contribute to the cellular generation of ATP. For this purpose, we formulated various fusogenic liposomes (Figure 1A) using DOPE and DOTAP in combination with fluorescently tagged lipids, such as NBD-PE and Rhod-PE. We loaded these liposomes with membrane-impermeable chromophores (e.g., calcein and DAPI) and PCr, and tested their interactions with the cell membrane and the delivery efficiency for these cargoes into the cellular and subcellular locations (Figure 1B,C). We hypothesized that liposomal membrane fusion would facilitate cytosolic delivery of the cargoes (calcein, DAPI, and PCr), which then further traverse to subcellular locations; for example, calcein traverses to intracellular calcium ions in the cytosol and DAPI to DNA in the nucleus. The PCr that is delivered into the cytosol via fusogenic liposomes contributes directly to ATP production.

Liposome preparation and physicochemical characterization. For successful liposome membrane–plasma membrane fusion, the liposomal lipid bilayer should exhibit a 3D lamellar phase structure with an inverted hexagonal structure [13,14,15,16,17,18]. This can be achieved by the incorporation of unsaturated fatty acid chain-containing lipids, such as DOTAP (cationic) and DOPE (nonionic). The inverted hexagonal structure in the 3D lamellar phase, combined with the cationic surface charge of DOTAP, serves as an intermediate driver of liposome–plasma membrane fusion [16,18]. Additionally, the incorporation of phospholipids containing aromatic molecules (e.g., BODIPY) with the DOPE/DOTAP mixture has been reported to promote an increase in the fusogenic capability of liposomes [6]. This is believed to be induced by local instability in the lipid bilayer due to the electrostatic interaction between the positively charged lipids and the highly polarizable π-electron system of the aromatic chromophores, which serves as an additional driving force to facilitate membrane fusion. In this study, we prepared four different liposomal formulations containing DOPE and DOTAP at a 1:1 ratio as their main lipids. These were spiked (final concentration 10%) with an aromatic fluorescent phospholipid 1,2-dioleoyl-sn-glycero-3-phosphoethanolamine-N-[lissamine Rhodamine B sulfonyl] (Rhod-PE) or 1,2-dioleoyl-sn-glycero-3-phosphoethanolamine-N-[7-nitro-2–1,3-benzoxadiazol-4-yl (NBD-PE). The addition of these lipids facilitates their fusogenecity and also allows for the tracking of their fusogenic efficiency in live cells using confocal imaging.

To test the fusion and cytosolic delivery of cargo-loaded liposomes, we prepared liposomes as follows: (1) DOPE/DOTAP/Rhod-PE (560 nm excitation (ex.)/583 nm emission (em.)) loaded with calcein (5 mM, 495 nm ex./515 nm em.) in the aqueous core (DOPE/DOTAP/Rhod-PE/Calcein); (2) DOPE/DOTAP/NBD-PE (460 nm ex./535 nm em.) loaded with DAPI only (2 mM, 359 nm ex./457 nm em.) (DOPE/DOTAP/NBD-PE/DAPI), (3) DOPE/DOTAP/NBD-PE loaded with PCr only (DOPE/DOTAP/NBD-PE/PCr); and (4) DOPE/DOTAP/NBD-PE loaded with both DAPI and PCr (DOPE/DOTAP/NBD-PE/PCr+DAPI). The crude liposomal formulations were extruded through polycarbonate membranes (200 nm) to remove unincorporated lipids and achieve a uniform size distribution and consistency among the different preparations. The liposomes were first characterized using dynamic light scattering (DLS) to assess their hydrodynamic diameter and zeta potential measurements to determine their surface charge. Figure 1 shows the results of these analyses. The data show that all the liposomal preparations exhibited a consistent hydrodynamic diameter (<200 nm) with a narrow polydispersity index (PDI, ≤0.20) across the different preparations. The concentration of all of the liposomal preparations was determined to be ~50–60 pM. The liposomes also showed highly positive (~+30 mV) surface charges across the different preparations, confirming the incorporation of the cationic DOTAP into the lipid membrane.

Characterization of fusogenic properties of liposomes. Next, we tested the liposomes for their fusogenic properties. Live human embryonic kidney cells (HEK 293T/17) were incubated with DOPE/DOTAP/Rhod-PE liposomes loaded with calcein, a cell-impermeable fluorophore, to determine the efficiency of liposomal fusion to the plasma membrane and the delivery of calcein to the cellular cytosol. We incubated the liposomes (3.0 pM) on the monolayer of HEK 293T/17 cells for 20 min at 37 °C, followed by washing to remove unfused liposomes. The Rhod-PE and calcein signals were imaged and quantified using confocal fluorescence microscopy. Figure 2A shows a representative confocal image where the Rhod-PE signal was evident on the plasma membrane (in ~50% of cells), indicating the successful fusion of the liposomal membrane with the plasma membrane. Further, in nearly all these same cells, the calcein signal was intracellular and homogeneous throughout the cell, indicating the successful delivery of the cell-impermeable calcein to the cytosol. It was important to assess the potential toxicity or damage to the plasma membrane caused by the positively charged lipid component, DOTAP, which has been shown in some cases to create transient pores in the plasma membrane [19,20]. To test this, we incubated free calcein (10 µM) on the cell monolayer that had already been incubated with the DOTAP-containing fusogenic liposomes, and images were collected at 1, 15, and 30 min time points (Figure 2B). Here, we observed that the intracellular green fluorescence signal remained unchanged while there was a steady, time-dependent increase in the green fluorescence signal in the extracellular spaces (Figure 2C). Over the 30 min observation period, we noted a ~6-fold increase in the extracellular green calcein signal compared to the intracellular calcein signal. This increase is most likely due to the incremental binding of calcein with the bottom of the dish, which is coated with fibronectin, as the incubation time progresses. These results confirm the successful fusion of the liposomes with the plasma membrane and the delivery of calcein to the cell interior while, at the same time, there was no deleterious effect on plasma membrane integrity.

We further demonstrated the functionality and compatibility of the fusogenic liposomes with a phospholipid bearing a different aromatic dye (NBD-PE, green). In this instance, we prepared DOPE/DOTAP/NBD-PE loaded with DAPI, a blue fluorescent nucleic acid counterstain. Analogous to the calcein experiments, DAPI is cell-impermeable and DNA staining was used to confirm successful liposome–plasma membrane fusion and intracellular cargo delivery. Figure 3A shows that when the DAPI-loaded DOPE/DOTAP/NBD-PE liposomes were incubated on HEK 293T/17 cells for 30 min, a robust, time-resolved green staining of the plasma membrane (NBD-PE) coupled with an increase in blue nuclear staining was obtained. This confirmed both liposome–plasma membrane fusion and the intracellular delivery of DAPI and subsequent binding to DNA in the nucleus. The time-resolved quantification of NBD staining of the plasma membrane (Figure 3B) and DAPI staining of the nuclei (Figure 3C) showed that while maximum green NBD fluorescence staining of the plasma membrane was achieved at ~15 min, the DAPI signal in the nucleus continued to increase over the 30 min imaging window. This is consistent with the continued influx of DAPI into the cytosol and eventually into the nucleus coupled with the increased fluorescence upon binding to the DNA. Overall, these results demonstrated the efficient delivery of DAPI from DOPE/DOTAP/NBD-PE to the cell interior.

Cellular delivery of PCr using fusogenic liposomes. Having established the utility of the fusogenic liposome system, we next sought to demonstrate the cytosolic delivery of PCr and the concomitant increase in cellular ATP levels. To do this, we synthesized liposomes containing PCr (DOPE/DOTAP/NBD-PE/PCr) and those without PCr (DOPE/DOTAP/NBD-PE). Figure 4A shows the time-resolved response of the fluorescent probe, BioTracker^TM^ ATP-Red, in HEK 293T/17 cells that were incubated with DOPE/DOTAP/NBD-PE/PCr liposomes. This probe increases in fluorescence in a quantitative manner upon binding to ATP. The fluorescence micrographs show the increase in cellular fluorescence (red) 30 min after incubation of the cells with the PCr-containing liposomes. Figure 4B shows the time-resolved quantitative comparison of fluorescence response in cells incubated with PCr-containing liposomes (blue trace) and cells incubated with cells incubated with non-PCr liposomes (red trace). We observed a 23% increase in ATP levels in PCr(+) cells over the 30 min window. While cells incubated with DOPE/DOTAP/NBD-PE or free PCr only showed no measurable increase in ATP-Red signal, cells incubated with DOPE/DOTAP/NBD-PE/PCr and DOPE/DOTAP/NBD-PE/PCr/DAPI showed increases of 23% and 17%, respectively, over the entire 30 min observation period (Figure 4C). That cells incubated with free PCr only showed no increase in ATP levels confirms that PCr does not readily enter cells on its own.

Cell viability. Finally, we determined the viability of HEK 293T/17 cells incubated with the various liposomal formulations. Viability was assessed using a cellular proliferation assay based on a tetrazolium compound (MTS) that, in viable cells, is converted into a blue formazan product that absorbs at 590 nm. Figure 5 shows the resulting cellular viability when cells were incubated for 30 min with liposomes up to the concentration of 6 pM. The cellular viability of cell incubated with DOPE/DOTAP/NBD-PE and DOPE/DOTAP/NBD-PE/PCr liposomes was >90% at a 6 pM liposome concentration. Cells incubated with DOPE/DOTAP/NBD-PE/PCr+DAPI liposomes showed a viability of ~75% at 6 pM. This is not surprising given the fact that the DAPI tracer binds to nuclear DNA and likely inhibits proliferation [21].

## 3. Materials and Methods

All chemicals, including the ATP imaging probe (BioTracker™ ATP-Red), were purchased from Millipore Sigma (St. Louis, MO, USA) and used as received unless otherwise noted. Phosphocreatine (PCr) was purchased from Santa Cruz Biotechnology, Inc. (Dallas, TX, USA). For liposome preparation, the lipids 1,2-Dioleoyl-sn-glycero-3-phosphoethanolamine (DOPE), 1,2-dioleoyl-3-trimethylammonium-propane, chloride salt (DOTAP), 1,2-dioleoyl-sn-glycero-3-phosphoethanolamine-N-[7-nitro-2–1,3-benzoxadiazol-4-yl (NBD-PE), and 1,2-dioleoyl-sn-glycero-3-phosphoethanolamine-N-[lissamine Rhodamine B sulfonyl] (Rh-PE) were purchased from Avanti Polar Lipids (Alabaster, AL). Dulbecco’s phosphate-buffered saline (DPBS), 4-(2-hydroxyethyl)-1-piperazineethanesulfonic acid (HEPES, 1M), and Dulbecco’s Modified Eagle Medium (DMEM) containing 25 mM HEPES (DMEM-HEPES) were purchased from Thermofisher Scientific (Waltham, MA, USA).

Liposome preparation and characterization. The liposomes were prepared using a lipid hydration method. In brief, lipid components were dissolved in chloroform and methanol (3:1, *v*/*v*) mixture in which the lipids (DOPE/DOTAP/NBD-PE or Rh-PE) were completely soluble at a ratio of 1/1/0.1 mol/mol. The lipid ratio for DOPE/DOTAP was chosen based on the previous study [6]. The solvent mixture was evaporated under a vacuum at 45 °C on a rotavapor to form a thin film of the lipid mixture in the round-bottom flask. Any trace of organic solvents was removed by keeping the flask under vacuum for an additional 12 h. The lipid thin film was hydrated in 20 mM 2-(4-(2-hydroxyethyl)-1-piperazinyl)-ethansulfonic acid (HEPES) buffer (pH 7.0) with a calcein (5 mM), DAPI (10 mM), PCr (10 mM), and PCr (10 mM)/DAPI (2 mM) mixture. The lipid suspension for each type of liposomal preparation was vortexed for 30 min and sequentially extruded through polycarbonate membranes of 1, 0.6, 0.4, and 0.2 µm pore size using a (Lipex Biomembranes Inc., Vancouver, BC, Canada). The extrusion was performed at 50 °C and repeated at least 3 times for each pore size. The liposomal preparations were purified using dialysis against the cellulose membrane (MWCO 10 kDa) and passed through a PD10 size exclusion column. The liposomes were characterized for their particle size, concentration, and surface charge and were stored at 4 °C and used for the study within 2 weeks. For the PCr delivery experiment, the liposomal formulations were used within 48 h of storing at 4 °C. The particle size distribution and concentration of the liposomes were measured via dynamic light scattering (DLS) of diluted (40× dilution) dispersion (in an HEPES buffer using a ZetaSizer NanoSeries equipped with a HeNe laser source (λ = 633 nm) (Malvern Instruments Ltd., Worcestershire, UK) and analyzed using Dispersion Technology Software (DTS, Malvern Instruments Ltd., Worcestershire, UK). Zeta potential was measured on a ZetaSizer NanoSeries equipped with a HeNe laser source (λ = 633 nm) (Malvern Instruments Ltd., Worcestershire, UK) and an avalanche photodiode for detection. For each analysis, at least four measurements were performed, and the data were reported as average values ± standard error of the mean (SEM).

Cell culture. HEK 293T/17 cells (ATCC CRL-11268, Manassas, VA, USA) were cultured as described previously [22], and all cellular delivery experiments were performed using cells between passages 5 and 15. Cells were seeded to 35 mm dishes with a 14 mm glass bottom insert (#1.0 cover glass, MatTek Corp., Ashland, MA, USA) at a density of ~7 × 10^4^ cells/mL (3 mL/well). Dishes were coated with fibronectin (10–20 µg/mL) in DPBS before adding the cell suspension. The cells were cultured for 24 h in standard incubation at 37 °C before confocal imaging.

Interaction of the fusogenic liposomes with the cells. In general, liposomal dispersions (3.0 pM) in DBPS were added directly to the cell monolayers in a microscope stage top imaging incubation chamber at 37 °C. To determine fusogenic efficiency in the plasma membrane and deliver the liposomal content to the cytosol, the cells were imaged (60× oil immersion objective) every 10 min for 2 h via differential interference contrast (DIC) and confocal laser scanning microscopy (CLSM). The confocal images were captured using a 405 nm diode laser, 488 nm argon laser, and 543 nm HeNe laser with fluorescence detection channels set to the filters of 425–475 nm (blue; for DAPI), 500–550 nm (green; for NBD-PE and calcein), and 570–620 nm (red; for ATP-Red) with dichroic mirrors at 405/488/561/640. Image analysis was performed by drawing regions of interest (ROI) in the fluorescent signal on the cells at different time points and data are represented as the percent of signal change compared to that at 0 min.

Cell viability assay. The cytotoxicity of the liposomal preparation was determined using the CellTiter 96^®^ Aqueous One Solution MTS Cell Proliferation Assay (Promega, Madison, WI, USA). This assay works based on the conversion of a tetrazolium substrate to a soluble formazan product by viable cells following a suitable proliferation period. Briefly, HeLa cells were seeded to 96-well tissue culture plates (~5 × 10^3^ cells/well in 100 µL media) and cultured for 24 h in standard incubation conditions. Next, 50 µL of DMEM-HEPES containing increasing concentrations of liposomes were added to the wells and incubated for 20 min at 37 °C. After the incubation, the materials were replaced with 100 µL of complete growth medium, and the cells were cultured for 72 h. After this proliferation period, 20 µL of the tetrazolium substrate was added to each well and incubated at 37 °C for 3 h to form the color product formazan. The absorbance of the formazan product was read at 490 nm and 590 nm (for the subtraction of nonspecific background absorbance) using a Tecan Spark Cyto600 (Tecan US Inc., Morrisville, NC, USA) microliter plate reader. The absorbance values (Abs) were converted to percent of cell viability by (Abs_treated_/Abs_control_) × 100, where Abs_treated_ and Abs_control_ represent the values of the background-subtracted absorbance measured from the liposome-treated cells and control cell (cultured in complete media only), and plotted as a function of material concentrations.

## 4. Conclusions

We demonstrated the potential of fusogenic liposomes as an efficient method for the intracellular delivery of the ATP precursor PCr to enhance the homeostasis of cellular energy. Our approach leveraged the unique properties of fusogenic liposomes to overcome the challenges associated with delivering PCr directly into the cytosol of cells, which is crucial for maintaining ATP levels, especially in conditions where creatine transporter CRT1 functionality is compromised. Fusogenic liposomes composed of a mixture of DOTAP, DOPE, and aromatic phospholipids were engineered to facilitate the direct fusion with the plasma membrane, allowing for the efficient delivery of their aqueous cargo, including PCr, into the cytosol. We demonstrated that these liposomes could successfully deliver membrane-impermeable chromophores, such as calcein and DAPI, as well as PCr, into the cellular cytosol. This was confirmed through confocal microscopy imaging, which showed that the cargoes were efficiently internalized without significant damage to the plasma membrane. PCr-loaded fusogenic liposomes effectively increased ATP levels within cells, as evidenced by the enhanced fluorescence of the ATP-sensing probe, ATP-Red. This increase in ATP levels was quantitatively significant, confirming that the fusogenic liposomes could successfully facilitate the cytosolic delivery of PCr and thereby contribute to ATP production. Notably, cells treated with fusogenic liposomes containing PCr exhibited a 23% increase in ATP levels, highlighting the efficacy of this delivery method in bypassing the limitations associated with direct PCr delivery from a bulk solution. The viability of cells treated with various liposome formulations showed that the fusogenic liposomes did not adversely affect cell viability at concentrations up to 6 pM, with cell viability remaining above 90% for most formulations. This suggests that the liposome-based delivery method is both effective and biocompatible, although the inclusion of additional tracers like DAPI may slightly impact cell proliferation. Overall, the successful implementation of fusogenic liposomes for the intracellular delivery of PCr aimed to address conditions related to ATP deficiency. This method holds promise for treating creatine transporter deficiencies and potentially other metabolic disorders by ensuring direct cytosolic delivery and avoiding extracellular hydrolysis.

## 5. Patents

A patent application resulting from the work reported in this manuscript has been filed with the United States Patent and Trademark Office.

## Data Availability

The original contributions presented in the study are included in the article, further inquiries can be directed to the corresponding author/s.

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
