# Peer review of "Fusogenic Liposomes for the Intracellular Delivery of Phosphocreatine"

_pharmaceuticals, 2024, doi:10.3390/ph17101351_

Round 1

Reviewer 1 Report

Comments and Suggestions for Authors

The authors report the formulation of fusogenic liposomes for intracellular delivery of phosphocreatine to maintain ATP levels inside cells. The idea is novel, and the hypothesis was proven via well-designed experiments. The manuscript addresses a knowledge gap related to the intracellular fate of cargo after administration within liposomes, and hence it would be of considerable interest to the readers. References are adequate and are mostly up to date. Only few minor comments to be addressed by the authors:

1- Figure 1, please report PDI values as mean and S.D.

2- Lines 317-319, please clarify how the solvent ratio and the ratio of lipid components were selected/optimized.

3- Line 325, please replace the word "suspension" with "dispersion", since suspension denotes micrometer-sized liquid.

4- Line 377, include the equation used to calculated cell viability.

Author Response

Overall Comment: The authors report the formulation of fusogenic liposomes for intracellular delivery of phosphocreatine to maintain ATP levels inside cells. The idea is novel, and the hypothesis was proven via well-designed experiments. The manuscript addresses a knowledge gap related to the intracellular fate of cargo after administration within liposomes, and hence it would be of considerable interest to the readers. References are adequate and are mostly up to date. Only few minor comments to be addressed by the authors:

Response: Thank you for the comments. The comment has been addressed and incorporated in the revised manuscript as appropriate.

Comment 1:- Figure 1, please report PDI values as mean and S.D.

Response: Mean PDI with S.D. has been added to Fig.1 in the revised manuscript.

Comment 2:- Lines 317-319, please clarify how the solvent ratio and the ratio of lipid components were selected/optimized.

Response: The revised manuscript includes details regarding the choice of solvent ratio and lipid composition ratio in the section “Liposome preparation and characterization” in Materials and Method.

Comment 3- Line 325, please replace the word "suspension" with "dispersion", since suspension denotes micrometer-sized liquid.

Response: The “suspension” has been replaced with “dispersion” in the relevant section in the revised manuscript.

Comment 4- Line 377, include the equation used to calculate cell viability.

Response: The revised manuscript includes the equation for the calculation of cell viability in the section “Cell Viability Assay” in Materials and Method.

Reviewer 2 Report

Comments and Suggestions for Authors

The article conducted by Prof. Delehanty and colleagues introduced a fusogenic liposomes for intracellular delivery of phosphocreatine. This technic is able to increase the ATP level inside the cell for various treatment. I believe the study is interesting and well written. However, I also feel more data is maybe need especially in formulation design to improve the quality of the paper. Below are my detailed comments and suggestions:

Major comments:

1. It will good to have micro and macro visualization of the fusogenic liposomes formulation. For example, SEM or TEM image can be used to demonstrate the morphology and micro structure of liposomes. A photo of the formulation is also recommended to show the suspension form without any precipitate.

2. I feel the stability of PCr in liposome is essential in this study. As we know, phosphocreatine can be converted into creatinine in seconds within cells to produce ATP, and it is not stable in aqueous solution. As PCr is loaded inside the aqueous core of liposome, I believe understanding the stability of PCr in liposome’s core is quite important in this study.

3. I have an issue in Figure 2B. Why the intensity of extracellular calcein increases over time? I believe the calcein used here was already fluorescent, which should reach maximum intensity outside the cell in a short period and then decreases because of quenching. Please explain why the intensity increases overtime.

4. In the article the author emphasis that fusogenic liposome can avoid endocytosis of the particle and prevent drug entrapment in lysosome. However, I did not any see any evidence in this part. I would be good if the author can demonstrate the delivery of liposomes containing dye along with lysotracker to visualize the distribution of the cargo.

Minor comments:

1. The figure caption of Figure 4C presented various color bar for no PCr, free PCr and etc. However, the figure itself is in black and white.

2. The last paragraph of Conclusions Section is not suitable to be located here. The paragraph is more suitable to be located in the discussion section and most content is duplicates of previous discussion and introduction. The Conclusions Section is also too long and usually no additional references will to be located in the Conclusion. I suggest to combine this last paragraph with previous discussion of introduction.

Author Response

Overall comment: The article conducted by Prof. Delehanty and colleagues introduced a fusogenic liposomes for intracellular delivery of phosphocreatine. This technic is able to increase the ATP level inside the cell for various treatment. I believe the study is interesting and well written. However, I also feel more data is maybe need especially in formulation design to improve the quality of the paper. Below are my detailed comments and suggestions:

Response: Thank you for the comments. We have addressed them and incorporated them into the revised manuscript as appropriate.

Major comments:

Comment 1. It will good to have micro and macro visualization of the fusogenic liposomes formulation. For example, SEM or TEM image can be used to demonstrate the morphology and micro structure of liposomes. A photo of the formulation is also recommended to show the suspension form without any precipitate.

Response: The liposomal formulations characterized by DLS and Zeta Potential analysis indicate their ~200 nm size range with positive (~+30 mV) surface charge and remain dispersed in an aqueous buffer at 4 deg. However, upon long-term storage (beyond 2 weeks), we observed a certain degree of aggregation and settling on the bottom, which is typical for particles with this size and charge range. The authors think the additional documentation for the liposomal preparations with TEM imaging or photographs would not produce much scientific value to the scope of this study.   

Comment 2. I feel the stability of PCr in liposome is essential in this study. As we know, phosphocreatine can be converted into creatinine in seconds within cells to produce ATP, and it is not stable in aqueous solution. As PCr is loaded inside the aqueous core of liposome, I believe understanding the stability of PCr in liposome’s core is quite important in this study.

Response: Thank you for this insightful comment. PCr is converted into creatine in seconds to produce ATP with the help of a cellular enzymatic system (creatine kinase). However, in extracellular aqueous buffer PCr is relatively stable, but its stability can be influenced by factors such as pH and temperature. At neutral pH (around 7.0), phosphocreatine can remain stable for several days, as suggested by the study reported in PLoS One. 2012; 7(8): e43178, where it is shown that the majority of PCr exists in the buffer/liposome up to 4 days of storing at 4°C. In our study, to maintain the stability of PCr, liposomes were formulated in a buffer with pH 7.0 and stored immediately at 4°C. The PCr-loaded liposomal formulations were used for live cell study within 48 hours. The revised manuscript includes this in the section “Liposome preparation and characterization” in Materials and Method.  

Comment 3. I have an issue in Figure 2B. Why the intensity of extracellular calcein increases over time? I believe the calcein used here was already fluorescent, which should reach maximum intensity outside the cell in a short period and then decreases because of quenching. Please explain why the intensity increases overtime.

Response: The extracellular 10 uM calcein used in this study is in carboxylate form, which is soluble in the extracellular media and shows strong fluorescent emission during the entire course of the experiment. During the imaging, an increase in extracellular calcium signal is related to the slow progress of its binding at the bottom of the dish, which was coated with fibronectin. The authors do not see reasons why the intensity of the calcein would decrease over time, given the overall volume of the buffer remains the same in the dish, and so the bulk concentration of the calcein in the buffer. The revised manuscript includes the following: “..Over the 30 min observation period, we noted a ~6-fold increase in the extracellular green calcein signal compared to the intracellular calcein signal. This increase is most likely due to the incremental binding of calcein on the bottom of the dish coated with fibronectin as the incubation time progresses. …” in the relevant section.

Comment 4. In the article the author emphasis that fusogenic liposome can avoid endocytosis of the particle and prevent drug entrapment in lysosome. However, I did not any see any evidence in this part. I would be good if the author can demonstrate the delivery of liposomes containing dye along with lysotracker to visualize the distribution of the cargo.

Response: As described in the Introduction, the DOPE/DOTAP lipid composition for the formulation of the fusogenic liposome is established and has been reported by various reports (e.g., Int. J. Mol. Sci. 2018, 19; J Photochem. Photobio B. 2017, 171, 43). Kolašinac et al. systematically decipher the DOPE/DOTAP liposome for their fusogenic properties with the inclusion of aromatic head lipids in the lipid mixture. With 50% DOTAP in the lipids mixture, the fusogenic capacity of the liposomes is reported to be about 96%, with no significant endocytosis. With ~50% of the DOTAP content in our liposomal formulations, we expect to have a similar degree of fusogenic properties. Our time-resolved confocal image capture upon incubation with the liposomes clearly indicates uniform plasma membrane labeling upon membrane fusion rather than punctate labeling (indication of endocytosis). This is further supported by the complete cytosolic labeling of calcein.

Minor comments:

Comment 1. The figure caption of Figure 4C presented various color bar for no PCr, free PCr and etc. However, the figure itself is in black and white.

Response: The caption for Figure 4C has been fixed in the revised manuscript.

Comment 2. The last paragraph of Conclusions Section is not suitable to be located here. The paragraph is more suitable to be located in the discussion section and most content is duplicates of previous discussion and introduction. The Conclusions Section is also too long and usually no additional references will to be located in the Conclusion. I suggest to combine this last paragraph with previous discussion of introduction.

Response: The Conclusion has been reformulated and shortened in the revised manuscript. 

Round 2

Reviewer 2 Report

Comments and Suggestions for Authors

The authors have answered all my queries. I do not have further comments. The article maybe accept while the editor and all the other reviewers agree.